# Ultrafast and persistent photoinduced phase transition at room temperature monitored by streaming powder diffraction

Marius Hervé [1,2], Gaël Privault[1,2], Elzbieta Trzop [1,2], Shintaro Akagi [3], Yves Watier[4], Serhane Zerdane[5], Ievgeniia Chaban[1,2], Ricardo G. Torres Ramírez [1,2], Celine Mariette [1,4], Alix Volte[4], Marco Cammarata[4], Matteo Levantino [4], Hiroko Tokoro [3,2] ✉, Shin-ichi Ohkoshi [6,2] ✉ & Eric Collet [1,2,7] ✉

Ultrafast photoinduced phase transitions at room temperature, driven by a single laser shot and persisting long after stimuli, represent emerging routes for ultrafast control over materials' properties. Time-resolved studies provide fundamental mechanistic insight into far-from-equilibrium electronic and structural dynamics. Here we study the photoinduced phase transformation of the $Rb_{0.94}Mn_{0.94}Co_{0.06}[Fe(CN)_6]_{0.98}$ material, designed to exhibit a 75 K wide thermal hysteresis around room temperature between $Mn^{III}Fe^{II}$ tetragonal and $Mn^{II}Fe^{III}$ cubic phases. We developed a specific powder sample streaming technique to monitor by ultrafast X-ray diffraction the structural and symmetry changes. We show that the photoinduced polarons expand the lattice, while the tetragonal-to-cubic photoinduced phase transition occurs within 100 ps above threshold fluence. These results are rationalized within the framework of the Landau theory of phase transition as an elastically-driven and cooperative process. We foresee broad applications of the streaming powder technique to study non-reversible and ultrafast dynamics.

The ability to control physical properties of materials on demand is an important challenge in science and technology. The use of light as external stimuli, spanning from UV to THz range, is very promising for contact-less, ultrafast and selective control[1–6]. Ultrashort laser pulses, containing a number of photons of the order of the number of active units within materials, open the way for ultrafast and cooperative switching of their physical properties through photoinduced polarons or lattice instabilities[7–9], with various applications for photonic devices or light-based technologies. For optically driven photonic devices, memories, or actuators, the photo-response must combine important characteristics such as room temperature switching, wide thermal regime of bistability, photoinduced states that can persist long after stimuli, single laser shot with threshold switching and ultrafast dynamics[10,11]. Molecular materials undergoing photoinduced phase transitions (PIPT) show various types of functionalities switchable under photoexcitation, such as conductivity, ferroelectricity, colour, etc.[12–20] Among the molecular materials, cyanide-bridged bimetallic assemblies are very attractive compounds exhibiting inter-metallic charge-transfer (CT), which allows for multi-functional properties that can be reversibly controlled by light, including magnetism, ionic

[1]Univ Rennes, CNRS, IPR (Institut de Physique de Rennes) - UMR 6251, 35000 Rennes, France. [2]CNRS, Univ Rennes, DYNACOM (Dynamical Control of Materials Laboratory) - IRL 2015, The University of Tokyo, 7-3-1 Hongo, Tokyo 113-0033, Japan. [3]Department of Materials Science, Faculty of Pure and Applied Sciences, University of Tsukuba, 1-1-1 Tennodai, Tsukuba, Ibaraki 305-8577, Japan. [4]ESRF – The European Synchrotron, 71 avenue des Martyrs, CS40220, 38043 Grenoble Cedex 9, Grenoble, France. [5]SwissFEL, Paul Scherrer Institut, Villigen, PSI, Switzerland. [6]Department of Chemistry, School of Science, The University of Tokyo, 7-3-1 Hongo, Bunkyo-ku, Tokyo 113-0033, Japan. [7]Institut universitaire de France (IUF), Paris, France. ✉e-mail: tokoro@ims.tsukuba.ac.jp; ohkoshi@chem.s.u-tokyo.ac.jp; eric.collet@univ-rennes.fr

conduction or optical properties such as rotation of light polarization[21–37].

Here we present a material of this class, $Rb_{0.94}Mn_{0.94}Co_{0.06}[Fe(CN)_6]_{0.98} \cdot 0.2H_2O$ (hereafter called $RbMn_{0.94}Co_{0.06}Fe$), which exhibits a wide thermal hysteresis centred at room temperature (Supplementary Methods 1) resulting from coupled CT and symmetry-

breaking (SB). We used a streaming powder diffraction technique developed to study its ultrafast and permanent PIPT, induced by a single laser shot at room temperature. The time-resolved X-ray diffraction data show that at low fluence the system remains in a tetragonal and expanded lattice distorted by CT polarons, which are responsible for reverse Jahn–Teller relaxation. Above threshold excitation, the lattice expansion due to photoinduced polarons switches the material from its low-temperature $Mn^{III}Fe^{II}$ tetragonal ground state to the persistent $Mn^{II}Fe^{III}$ cubic phase, on the 100 ps timescale. The multiscale nature of the process underlines the key role of elastic interactions in the threshold and cooperative phase transition.

## Results

### Room-temperature charge-transfer bistability

The synthesized $RbMn_{0.94}Co_{0.06}Fe$ material exhibits a CT-based phase transition similar to the one of the pure $RbMnFe(CN)_6$ system and derivatives, as characterized by magnetic, X-ray diffraction and infrared measurements (Supplementary Methods 1)[38]. Figure 1 shows the thermal hysteresis associated with the $Mn^{II}Fe^{III} \leftrightarrow Mn^{III}Fe^{II}$ bistability, monitored through the thermal dependence of the $\chi_M T$ product (molar magnetic susceptibility $\chi_M$ and temperature $T$). The $\chi_M T$ value at low temperature (LT) is characteristic of the $Mn^{III}(S=2)Fe^{II}(S=0)$ state. Upon warming it increases around $T_\uparrow = 328$ K to reach a value characteristic of the $Mn^{II}(S=5/2)Fe^{III}(S=1/2)$ high temperature (HT) state. Upon cooling the $\chi_M T$ value suddenly drops back to LT value around $T_\downarrow = 253$ K, resulting in a 75 K wide hysteresis. Compared to the pure $RbMnFe(CN)_6$ material, for which the thermal hysteresis is centred at 267 K ($T_\downarrow = 231$ K and $T_\uparrow = 304$ K)[38], Co doping allows chemical tuning of the bistability regime, centred at room temperature (290.5 K) for $RbMn_{0.94}Co_{0.06}Fe$. The infrared spectra in the LT and HT phases are characteristic of the $Mn^{III}Fe^{II}$ and $Mn^{II}Fe^{III}$ states. Powder X-ray diffraction data show important changes of lattice parameters between the HT cubic space group $F\bar{4}3m$ ($a_{HT} = 10.5495(6)$ Å) and the LT tetragonal space group $F\bar{4}2m$ ($a_{LT} = 10.0051(11)$ Å, $c_{LT} = 10.4744(16)$ Å). This volume contraction between the cubic HT ($V_{HT}$) and tetragonal LT ($V_{LT}$) phases corresponds to a large volume strain: $v_s = \frac{V_{LT} - V_{HT}}{V_{HT}} \simeq -0.1$. The use of the non-conventional $F\bar{4}2m$ LT space group allows for direct description of the symmetry breaking between the two phases, which is due to the collective Jahn–Teller distortion along the $z$-axis, stabilizing the $Mn^{III}$ LT state. As introduced for $RbMnFe$[38], the cubic → tetragonal symmetry-breaking ferroelastic distortion of the crystalline lattice, due to the loss of the three-fold rotational symmetry, is characterized by its amplitude:

$$\eta(T) = \frac{2}{\sqrt{3}}\left(\frac{c_{LT}(T) - a_{LT}(T)}{a_{HT}}\right) \quad (1)$$

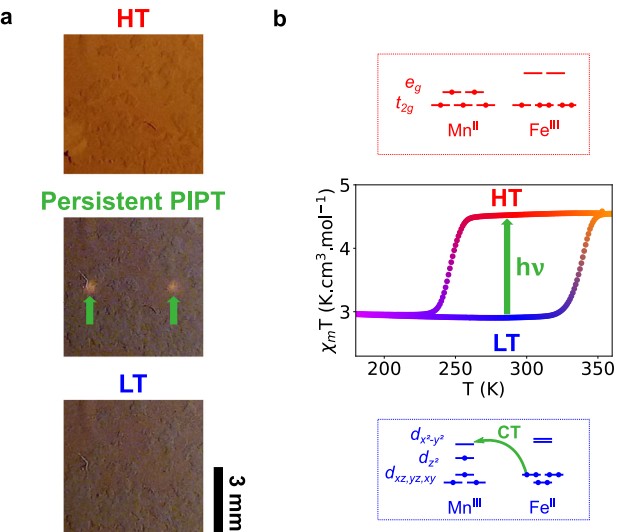

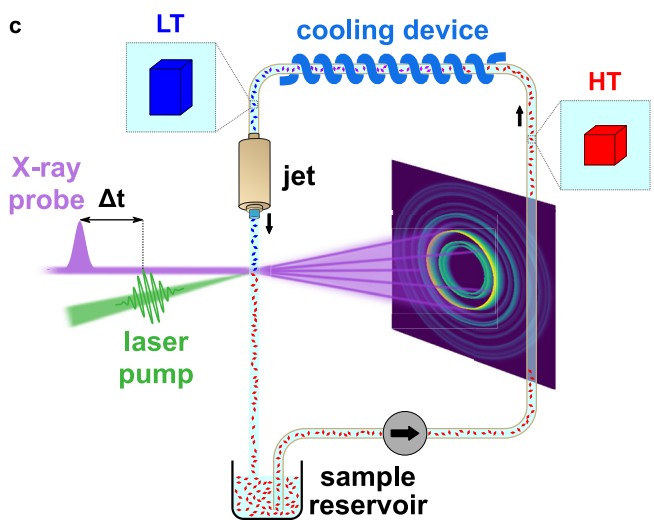

**Fig. 1 | Charge-transfer bistability at room temperature. a** Pictures of $6 \times 6$ mm² $RbMn_{0.94}Co_{0.06}Fe$ thin films taken at room temperature, showing the colour change between the $Mn^{III}(S=2)Fe^{II}(S=0)$ low-temperature (LT) and $Mn^{II}(S=5/2)$ $Fe^{III}(S=1/2)$ high-temperature (HT) phases. Starting from the LT phase, a persistent photoinduced phase transition (PIPT) is induced by laser excitation (120 W cm⁻², 0.2 mm² spot size, 3 s irradiation), as characterized through colour change at laser spot positions (green arrows). Subsequent heating of the irradiated film yields global colour change over the whole film in the HT phase. **b** Thermal hysteresis between the HT and LT phases, characterized through the $\chi_M T$ vs $T$ plot, with the schematic representation of the electronic configurations showing the charge transfer (CT). **c** Streaming powder diffraction technique used for time-resolved measurements, performed on sub-μm crystals of $RbMn_{0.94}Co_{0.06}Fe$ dispersed in solution and streamed through a liquid jet. The X-ray beam probes the sample at a time delay $\Delta t$ after laser pumping within the hysteresis, which may eventually convert crystals from LT (blue) to permanent HT phase (red). The suspension in the reservoir circulates through a cooling device (230 K $< T_\downarrow$), bringing crystals back to the ground LT phase, before injecting them anew at room temperature in the pump-probe beams.

In the framework of the Landau theory of phase transitions[38], we have shown that it is also necessary to use a non-symmetry-breaking parameter $q$ to monitor the CT conversion:

$$q = \frac{N_{Mn^{II}Fe^{III}} - N_{Mn^{III}Fe^{II}}}{N_{Mn^{II}Fe^{III}} + N_{Mn^{III}Fe^{II}}} \quad (2)$$

where $N_{Mn^{II}Fe^{III}}$ and $N_{Mn^{III}Fe^{II}}$ denote the number of sites in each CT states. We have shown for similar materials that, during thermal equilibrium phase transitions, the elastic couplings of both parameters $q$ and $\eta$ to $v_s$ are responsible for their simultaneous changes from the LT $Mn^{III}Fe^{II}$ tetragonal ($q = -1$, $\eta \neq 0$) to the HT $Mn^{II}Fe^{III}$ cubic ($q = 1$, $\eta = 0$) phases and for widening the thermal hysteresis[38]. As shown in Fig. 1a, the change of electronic state is associated with colour change between $Mn^{III}Fe^{II}$ tetragonal and $Mn^{II}Fe^{III}$ cubic states, both stable at room temperature. The LT-to-HT phase transition can also be induced by laser irradiation at room temperature within the thermal hysteresis.

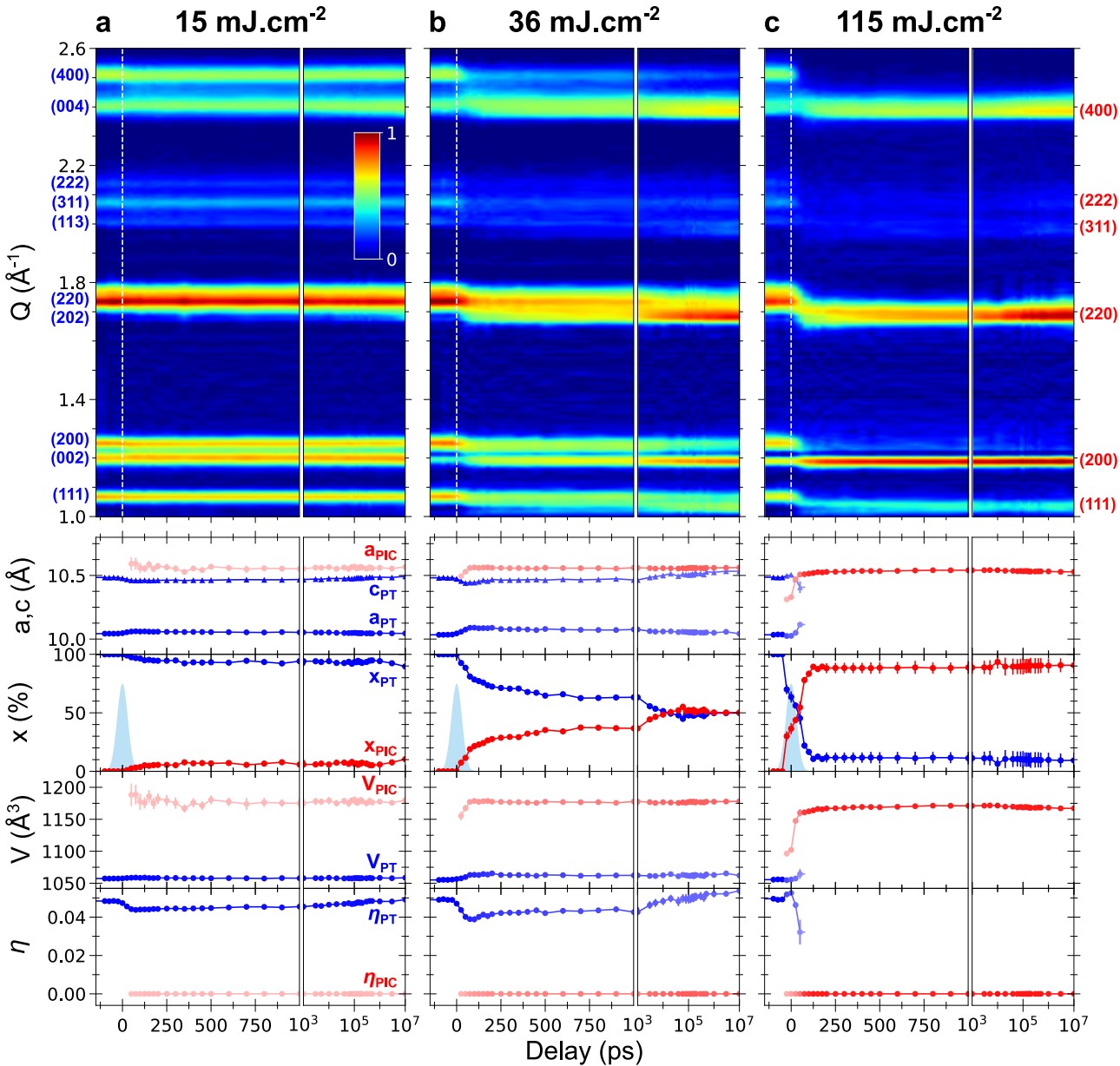

**Fig. 2 | Time-resolved X-ray diffraction.** Diffracted signal along the scattering vector Q (top) shown with colour-coded intensity and measured at room temperature for low (**a**, 15 mJ cm⁻²), medium (**b**, 36 mJ cm⁻²) and high (**c**, 115 mJ cm⁻²) pumping fluences. The lattice parameters ($a$, $c$, $V$, $\eta$) and the fractions of photoexcited tetragonal (PT, in blue) and photoinduced cubic (PIC, in red) phases are obtained from Rietveld refinement (bottom), and the associated error bars represent the standard deviation of the Rietveld fit for each delay. The time resolution (instrument response function) is shown by the blue Gaussian.

As shown in Fig. 1a, the characteristic colour change occurs at the laser spots, leading to a permanent photoinduced HT phase. In the following, we are interested in understanding the ultrafast out-of-equilibrium dynamics induced by a picosecond laser pulse and driving the $Mn^{III}Fe^{II}$ tetragonal → $Mn^{II}Fe^{III}$ cubic PIPT within the thermal hysteresis regime of bistability.

### Time-resolved crystallography with powder streaming

Understanding the underlying physical mechanisms driving PIPT is mandatory for controlling by light the peculiar properties of these materials. Various ultrafast studies, making use of different techniques, were performed on cyanide-bridged bimetallic assemblies, such as Fe–Fe, Co–Fe, Mn–Fe or V–Cr, and focused on the ultrafast formation of local and transient CT states[32,34,35,39–43]. In the present work we investigate by streaming powder diffraction (Fig. 1c) the out-of-equilibrium structural dynamics of a macroscopic PIPT in RbMn$_{0.94}$Co$_{0.06}$Fe, resulting from a single laser shot excitation within the thermal hysteresis. Conventional time-resolved diffraction techniques consist of exciting a system with a pump pulse and monitoring the change in the X-ray diffraction signal at a given delay $\Delta t$[6,13,44–48]. Then, the system rapidly recovers the ground state to be excited anew so that stroboscopic pump-probe experiments can be performed at high repetition rates (>1 Hz) on the same sample. However, this technique cannot be used to study non-reversible phenomena, such as photoswitching within thermal hysteresis. In parallel, serial crystallography, where time-resolved diffraction is performed on fresh single crystals for each laser shot, was largely applied in biology and recently developed for small-unit-cell systems[49–52]. Regrettably, it cannot be used in the present case because of (i) the limited number of accessible Bragg peaks, given the small cell parameters and high symmetry of RbMn$_{0.94}$Co$_{0.06}$Fe and ii) the small size of the crystals giving rise to weak single crystal diffraction. Therefore, we developed a streaming

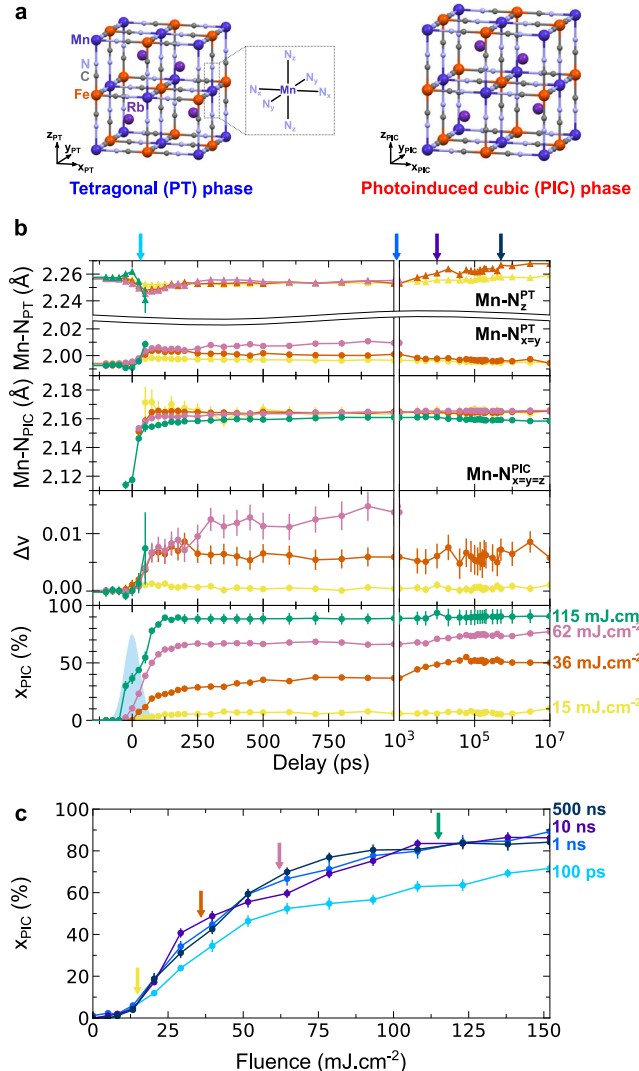

**Fig. 3 | Structural reorganizations towards photoinduced cubic phase. a** Room temperature photoinduced phase transition from the photoexcited tetragonal (PT) phase to the permanent photoinduced cubic (PIC) phase. **b** Temporal evolution of the Mn–N bond lengths in the PT and PIC phases (Mn–N$_{x=y}^{PT}$, Mn–N$_z^{PT}$, Mn–N$_{x=y=z}^{PIC}$), the relative volume expansion ($\triangle v$) of the PT lattice, and the fraction of PIC phase ($x_{PIC}$) for different laser fluences (yellow: 15 mJ cm$^{-2}$, orange: 36 mJ cm$^{-2}$, magenta: 62 mJ cm$^{-2}$, green: 115 mJ cm$^{-2}$). Reverse Jahn–Teller distortion, $\triangle v$ and $x_{PIC}$ increase with laser fluence. The time resolution is shown by the blue Gaussian. **c** Fluence dependence of $x_{PIC}$ for different time delays (light blue: 100 ps, blue: 1 ns, purple: 10 ns, dark blue: 500 ns), evidencing threshold fluence for PIPT. The fluences and delays colour codes (arrows and curves) are corresponding between **b** and **c**, and the associated error bars represent the standard deviation of the Rietveld fit for each delay and fluence.

powder diffraction technique (Fig. 1c, Supplementary Methods 2 and 3), which allows for studying out-of-equilibrium photoswitching dynamics inside thermal hysteresis by time-resolved X-ray diffraction (TR-XRD). We used the optical pump−X-ray probe configuration at the ID09 beamline of the European Synchrotron Radiation Facility (ESRF)[53] with ~35 ps experimental time resolution. The sample was excited with 1 ps laser pulses. The photoinduced Mn$^{III}$Fe$^{II}$ → Mn$^{II}$Fe$^{III}$ CT occurs within 200 fs over a broad excitation spectral range including Fe to Mn CT band (around 500 nm) and Mn-centred band (around 650 nm)[30]. We excited the sample at 650 nm, which is a good compromise between excitation efficiency and penetration depth (of the order of sample thickness). The typical size of the crystals (0.9 ± 0.3 μm), smaller than the laser pump penetration depth (1.9 μm)[54], ensures a

relatively homogeneous excitation at the scale of individual crystals dispersed in ethanol (transparent at 650 nm). Streaming powder diffraction experiments consist of performing TR-XRD measurements on a collection of micro-crystals dispersed in solution and streamed through a liquid jet, where they interact with the optical pump and delayed X-ray probe beams, at 1 kHz repetition rate in the present case (Fig. 1c). The overall global fraction of LT crystals exposed to laser shot in the streamed suspension is small. In addition, this suspension circulates through a cooling loop set below $T_\downarrow$, which allows to bring back to the ground LT phase the crystals eventually photoconverted. This ensures that each powder diffraction pattern is measured from a fresh batch of crystals initially in the ground LT state. This closed-loop circulation allows for long data acquisition time, often required in TR-XRD measurements. Both the HT and LT phases were measured at the room temperature of the jet. The lattice parameters of the HT ($a_{HT} = 10.544(2)$ Å) and the LT ($a_{LT} = 10.023(2)$ Å, $c_{LT} = 10.461(2)$ Å) phases, obtained from the Rietveld refinements in the streaming powder diffraction set-up, agree with conventional X-ray powder diffraction data (Supplementary Fig. 7), which shows the quality of the data obtained with streaming powder diffraction. Hereafter we present time-resolved streaming powder diffraction results, which made it possible to track the out-of-equilibrium structural dynamics towards the persistent PIPT, within the thermal hysteresis at room temperature.

## Out-of-equilibrium structural dynamics

Figure 2 shows the evolution of the diffraction pattern of RbMn$_{0.94}$Co$_{0.06}$Fe crystals, in the initial LT phase, and measured at room temperature as a function of pump-probe delay $\Delta t$. We used a Rietveld refinement (Supplementary Methods 4) to track the time evolution of the lattice parameters, phase fractions, and bond lengths for the different phases coming into play. At negative delays, the diffraction pattern corresponds to the initial Mn$^{III}$Fe$^{II}$ LT phase (F$\overline{4}$2m space group), where the characteristic Jahn-Teller ferroelastic distortion translates through the splitting of the ($h00$) and ($00h$) Bragg peaks for example. At positive delays, we had to consider two time-dependent phases: a tetragonal phase that describes the evolution of the original phase, called hereafter the photoexcited tetragonal (PT) phase, and a photoinduced cubic (PIC) phase, whose crystalline structure is similar to the HT phase.

Figure 2a shows that for low pumping fluence (15 mJ cm$^{-2}$) the initial LT Bragg peaks shift with delay. The structural Rietveld refinements of the diffraction patterns provide the evolution of the lattice parameters and of the ferroelastic distortion $\eta(t) \propto (c(t) - a(t))/a_{HT}$ for each individual delay $t$. The lattice parameters of the PT lattice evolve within 100 ps: $c_{PT}$ decreases while $a_{PT} = b_{PT}$ increase, resulting in a global volume increase (1 Å$^3$) and decrease of $\eta$. The lattice parameters relax back to initial equilibrium values within 10 μs. Similar lattice distortions were observed at low excitation density under photoexcitation outside thermal hysteresis (below $T_\downarrow$) of the Rb$_{0.94}$Mn[Fe(CN)$_6$]·2.5H$_2$O sample[33]. Since RbMnFe and derivative materials exhibit no thermal expansion[55], it was shown that such long-range lattice deformation results from the structural trapping of photoinduced Mn$^{II}$Fe$^{III}$ CT state of higher volume, which forms within 200 fs and decays within 10 μs[34]. At low fluence, the dynamical lattice response therefore includes two components: (i) an impulsive lattice response driven by collective reverse Jahn–Teller reorganizations, elongating $a_{PT}$ and shrinking $c_{PT}$, and (ii) a volume expansion induced by the long-lived photo-induced CT Mn$^{II}$Fe$^{III}$ small polarons, populating antibonding Mn $d(x^2 - y^2)$ orbitals. The Rietveld analysis also reveals that a small fraction of photoinduced cubic (PIC) phase appears on the 100 ps timescale ($x_{PIC} \simeq 10\%$).

The dynamical behaviour drastically changes when the laser fluence is increased to 115 mJ cm$^{-2}$ (Fig. 2c): the diffraction pattern of the initial LT phase almost disappears within 75 ps, while the Bragg peaks of the PIC phase form, characterized by the merging of ($h00$) and

($00h$) Bragg peaks. Here again, Rietveld refinements enable further understanding of the underlying dynamics at high fluence. After photoexcitation, the lattice parameters $a_{PT}$ and $c_{PT}$ converge within 100 ps towards the cubic lattice parameters. This is due to a larger and global reverse Jahn-Teller distortion, characterized by the fast drop of $\eta$ for the tetragonal lattice. In addition, the fraction of photoexcited tetragonal phase $x_{PT}$ also drops within 75 ps, while the fraction $x_{PIC}$ of the PIC phase grows rapidly and approaches 100%. Overall, Rietveld analysis shows that high laser fluence drives within 100 ps a global photoinduced $Mn^{III}Fe^{II}$ tetragonal → $Mn^{II}Fe^{III}$ cubic phase transition inside the thermal hysteresis and at room temperature. Compared to $V_{LT} = 1055\,\text{Å}^3$, the volume of the PIC phase increases from $V_{PIC} = 1096(5)\,\text{Å}^3$ within the experimental time resolution to $V_{PIC} = 1167(2)\,\text{Å}^3$ once conversion is complete, which is similar to the volume $V_{HT}$ of the thermodynamically stable HT phase. The remaining fraction of the tetragonal phase is very likely due to the limited laser excitation of crystals in the jet. Interestingly, measurements at intermediate fluence (36 mJ cm$^{-2}$, Fig. 2b) show a halfway response between low and high excitation densities with intensity decrease and shifts for the PT Bragg peaks and appearance of the PIC ones. Rietveld refinement indicates almost 50% conversion from LT to PIC phases at 36 mJ cm$^{-2}$, while the fraction of photoexcited sample remaining in the tetragonal lattice exhibits larger reverse Jahn−Teller distortion compared to low fluence. Here again, the tetragonal lattice recovers its initial structure within 10 μs as CT polarons decay, while the fraction of PIC phase is persistent.

## Discussion

Figure 3a shows the crystalline structure of the PIC phase, measured at 1 ns and 36 mJ cm$^{-2}$. The structural data of the PIC phase shown in Supplementary Table 2 are characteristic of the HT $Mn^{II}Fe^{III}$ phase, confirming the occurrence of photoinduced $Mn^{III}Fe^{II}$ tetragonal → $Mn^{II}Fe^{III}$ cubic conversion towards the HT phase. To get deeper insights into the PIPT mechanism, we monitored the photoconversion dynamics as a function of laser fluence, through the temporal evolution of the lattice parameters, $x_{PIC}(t)$ and the relative volume expansion $\Delta v(t) = \frac{V_{PT}(t) - V_{LT}}{V_{HT}}$ of the photoexcited tetragonal lattice at different fluences (Fig. 3b), and through the fluence dependence of $x_{PIC}(t)$ (Fig. 3c). Again, these parameters change within 100 ps. This timescale matches the ratio between crystals' radius (450 nm) and sound velocity (c = 4300 m s$^{-1}$ [56]), which is characteristic of an elastically-driven process[33,57]. Indeed, since RbMnFe derivatives exhibit no thermal expansion[55], the initial lattice expansion or ferroelastic distortion cannot be explained by a simple laser heating of the lattice. As can be seen on the evolution of $x_{PIC}(t)$, a second and weaker additional conversion step is observed for intermediate fluences around 10 ns, and is very likely due to slower heat diffusion as observed in spin-crossover materials[57–59]. Figure 3c shows the fluence-dependent evolution of $x_{PIC}$ at delays of 100 ps, 1, 10 and 500 ns. It clearly evidences a threshold fluence ($\simeq$10 mJ cm$^{-2}$) above which the $Mn^{III}Fe^{II}$ tetragonal → $Mn^{II}Fe^{III}$ cubic PIPT is fully established within less than 1 ns.

For fluences below the threshold, Fig. 3b shows that the less bonding photoinduced CT $Mn^{II}Fe^{III}$ state is responsible for Mn-N bond elongations, which translates into the volume expansion $\Delta v$ of the photoexcited tetragonal lattice. In this polaronic regime, $\Delta v$ increases with excitation density, while the lattice of the photoexcited crystals retains tetragonal symmetry.

On the other hand, anisotropic lattice distortions in this regime are explained by the anisotropic reorganization around the Mn−N$_6$ octahedron, as evidenced in Fig. 3b and S7: the structural refinements show elongation of the Mn−N$_{x,y}$ bonds and a contraction of the Mn−N$_z$ ones. This is the direct microscopic evidence of the structural reorganization around the Mn sites from D$_{4h}$ in $Mn^{III}Fe^{II}$ configuration towards O$_h$ for $Mn^{II}Fe^{III}$. This structural reorganization is characteristic of the reverse Jahn−Teller distortion associated with the photoinduced

CT polarons[33,34], which are localized at the level of a single Mn−N−C−Fe unit as previously characterized by time-resolved IR spectroscopy[43]. This collective local structural dynamics, driven by the photoinduced CT $Mn^{II}Fe^{III}$ polaron, is the driving force of the macroscopic anisotropic lattice deformations: $a_{PT}$ elongates, $c_{PT}$ contracts and therefore $\eta$ decreases.

Above 10 mJ cm$^{-2}$, the PIC phase forms within 100 ps and the conversion rate saturates with an almost complete conversion above 100 mJ cm$^{-2}$. This corresponds to the PIPT regime, with a persistent fraction of the PIC phase. Such a threshold response and fluence dependence are characteristic of cooperative PIPT[17,57–59]. Together, the ultrafast nature of the phase transformation and its non-linear fluence dependence clearly show that the PIC phase forms through a nucleation process, where a critical fraction of photoinduced CT $Mn^{II}Fe^{III}$ polarons can drive macroscopic PIPT. Since is hard to estimate precisely the number of absorbed photons, we estimate the percentage of photoexcited units by considering that quantum efficiency is of the order of unity and volume changes linearly with CT fraction[33,34]. Compared to the relative volume expansion from LT to HT phases ($\Delta v \sim 0.1$) the relative photoexpansion of the PT phase at 62 mJ cm$^{-2}$ ($\Delta v \sim 0.01$, in Fig. 3b) allows us to estimate that 10% of the MnFe units are photoexcited around the excitation threshold.

For each regime, the mechanism at play is deeply related to the evolution of the elastic deformation of the crystals, in particular the volume expansion and the release of the Jahn−Teller distortion. Below threshold excitation, the lattice pressure brings within 10 μs the higher volume photoinduced $Mn^{II}Fe^{III}$ polarons back to their lower volume ground $Mn^{III}Fe^{II}$ state, as characterized by the Mn−N bonds evolution (Fig. 3b). Above threshold, CT polarons drive larger volume expansion, which favours the cubic $Mn^{II}Fe^{III}$ phase of higher volume. Therefore, the overall threshold photoresponse results from the competition between pushing forces of the lattice on the CT polarons, driving their relaxation towards lower volume ground state, and pulling forces of the expanded lattice on ground $Mn^{III}Fe^{II}$ sites, favouring switching towards higher volume $Mn^{II}Fe^{III}$ state.

The role of elastic cooperativity in the present dynamics can be understood within the framework of the Landau theory of phase transition, which we have previously used to describe thermal equilibrium phase transitions in materials exhibiting coupled change of electronic state and symmetry, such as RbMnFe[38,60,61]. Indeed, the relevant Landau potential should include both the order parameter $q$ monitoring CT and the symmetry-breaking order parameter $\eta$ (Supplementary Discussion 1). These parameters are coupled to the volume strain $v_s$. Inside the thermal hysteresis, both the $Mn^{II}Fe^{III}$ cubic phase ($\eta_{HT} = 0$, $q = 1$) and $Mn^{III}Fe^{II}$ tetragonal phase ($\eta_{LT} \neq 0$, $q = -1$) are stable. Hereafter we focus our attention on the more usual Landau potential corresponding to the cubic → tetragonal symmetry breaking phase transition[62–64]:

$$G(\eta) = \frac{1}{2} a' \eta^2 + \frac{1}{3} b \eta^3 + \frac{1}{4} c \eta^4 \qquad (3)$$

where the renormalized coefficient $a' = a + 2\lambda_\eta v_s$ includes both the usual temperature-induced symmetry-breaking term ($a = a_0(T - T_{SB})$) and the elastic coupling of the ferroelastic distortion to $v_s$ ($v_s \eta^2 < 0$) where $v_s < 0$ stabilizes the tetragonal phase[65,66]. $a$ changes sign at the symmetry-breaking temperature $T_{SB}$, which stabilizes $\eta = 0$ above $T_{SB}$ and $\eta \neq 0$ below, while the symmetry-allowed $\eta^3$ term limits the $F\bar{4}3m → F\bar{4}2m$ ferroelastic transition to first-order. Inside the thermal hysteresis, both the high-volume cubic phase ($\eta_{HT} = 0$, $v_s = 0$) and low-volume tetragonal phase ($\eta_{LT} \neq 0$, $v_s < 0$) are stable, due to the elastic coupling to $v_s$ of the ferroelastic distortion $\eta$. Figure 4 shows the $G(\eta)$ curve obtained from Eq. (3) when cubic and tetragonal phases are equally stable (left panel). To describe the lattice response to photoexcitation, we should consider the results from the

### a   low fluence: polaronic regime

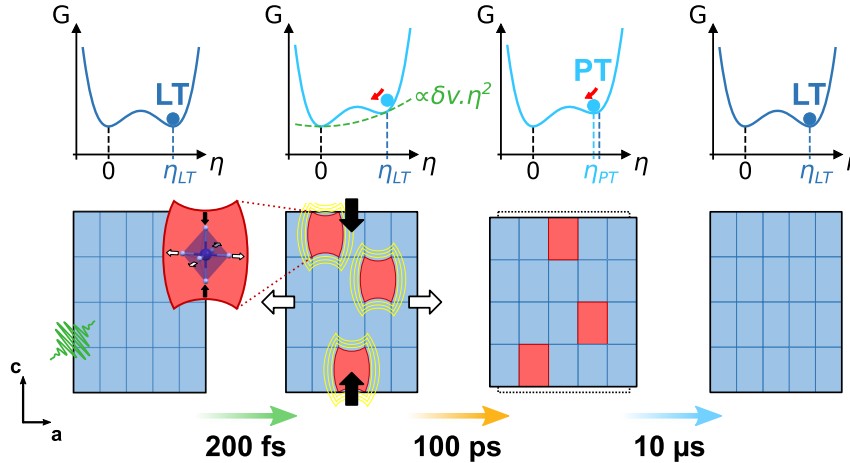

### b   high fluence: PIPT regime

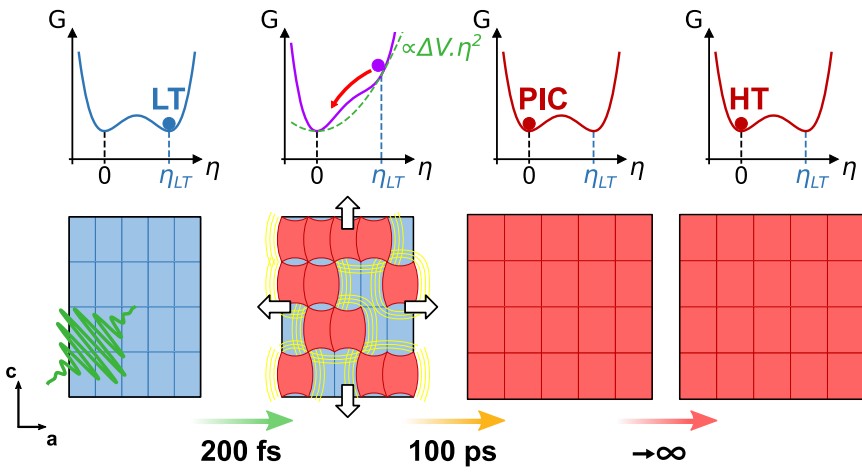

**Fig. 4 | From local charge-transfer polarons to macroscopic photoinduced phase transition driven by lattice expansion. a** Transient polaronic regime at low fluence inside thermal hysteresis, where $G(\eta)$ corresponds to equally stable low-temperature (LT, $\eta_{LT} \neq 0$) and high-temperature (HT, $\eta_{HT} = 0$) phases (dark blue curve). Excitation of the tetragonal $Mn^{III}Fe^{II}$ ground state results in charge-transfer (CT) polarons (represented as red sites in the crystal) responsible for a weak volume expansion $\delta v$. The additional elastic coupling cost $\propto \delta v \eta^2$ (dashed green curve) reduces the amplitude of the ferroelastic distortion, which equilibrates on elastic timescale (100 ps) to $\eta_{PT}$ in the photoexcited tetragonal (PT) lattice (light

blue curve). As CT polarons, and therefore $\delta v$, decay within 10 μs, the system relaxes towards the initial ground $Mn^{III}Fe^{II}$ state (dark blue curve). **b** Persistent photoinduced phase transition (PIPT) regime at high fluence, where the larger volume expansion $\triangle V$ strongly destabilizes the tetragonal lattice within 100 ps (purple curve), converting crystals towards the photoinduced cubic phase (PIC, $\eta_{PIC} = 0$, red curve). The relaxed photoinduced cubic $Mn^{II}Fe^{III}$ phase is then persistent at room temperature, due to the energy barrier with the initial tetragonal $Mn^{III}Fe^{II}$ phase, matching the HT phase (red curve).

time-resolved measurements discussed above, which bring key information on the mechanism behind macroscopic and ultrafast PIPT within the thermal hysteresis. The photoinduced $Mn^{II}Fe^{III}$ CT polarons, forming within 200 fs and living up to 10 μs[34], are sources of anisotropic elastic deformation waves, which propagate at the crystal scale and equilibrate with the lattice on the elastic timescale (100 ps). Compared to the initial LT volume strain $v_s$, the volume strain $v_s^*$ of the photoexcited lattice is modified by the small polarons resulting from photoinduced CT ($\Delta q$) and responsible for volume expansion $\Delta v$:

$$v_s^* = v_s + \Delta v \qquad (4)$$

The associated photoinduced volume expansion destabilizes the tetragonal lattice, through the elastic coupling cost term $\Delta v \eta^2 > 0$ (with $\Delta v > 0$). Figure 4a depicts the mechanism at play in the transient

polaronic regime at low fluence: the weak volume expansion $\delta v$ reduces the ferroelastic distortion in the transiently equilibrated PT state ($\eta_{PT} < \eta_{LT}$), which remains tetragonal due to the energy barrier towards the cubic phase. As the photoinduced CT polarons decay within 10 μs, the system relaxes back to the ground LT state with initial volume strain $v_s$ and ferroelastic distortion $\eta_{LT}$. In the persistent PIPT regime above threshold fluence (Fig. 4b), the high laser fluence induces larger volume expansion $\Delta V$ and therefore larger elastic coupling cost ( $\Delta V \eta^2$), that fully destabilizes the tetragonal lattice. As a result, full conversion towards the high-volume cubic symmetry ($\eta = 0$) occurs within the elastic timescale. The system then relaxes towards the equilibrium $Mn^{II}Fe^{III}$ cubic phase, corresponding to the HT phase. As the energy barrier with the tetragonal $Mn^{III}Fe^{II}$ phase appears anew after global relaxation, this photoinduced HT phase is persistent at room temperature, and more generally over the wide thermal hysteretic regime of bistability.

The present results demonstrate the possibility to induce by a single laser shot an ultrafast PIPT at room temperature, in a functional material designed to exhibit a wide regime of thermal bistability. Compared to previous single laser shot photoswitching, observed in another RbMnFe derivative at low temperature by static X-ray diffraction studies[67], our time-resolved analysis also brings key information on the out-of-equilibrium structural and symmetry dynamics. Indeed, these long-range structural reorganizations can only be directly monitored by ultrafast diffraction techniques. The cooperative tetragonal $Mn^{III}Fe^{II} \rightarrow$ cubic $Mn^{II}Fe^{III}$ photoinduced phase transition, which occurs above threshold excitation fluence, is persistent and is driven by elastic cooperativity. The energy barrier in the thermodynamic potential between the tetragonal $Mn^{III}Fe^{II}$ and cubic $Mn^{II}Fe^{III}$ phases has two important roles. On the one hand, it is responsible for the threshold fluence necessary to induce the cubic phase, as a critical volume expansion is required to destabilize the tetragonal lattice. On the other hand, this volume expansion stabilizes the photoinduced cubic $Mn^{II}Fe^{III}$ phase, which is persistent over a wide range of bistability. The streaming powder diffraction set-up that we developed made it possible to track structural dynamics at microscopic (Mn–N bonds) and macroscopic (lattice parameters) scales through the non-reversible symmetry and volume changes associated with the photoinduced tetragonal $Mn^{III}Fe^{II} \rightarrow$ cubic $Mn^{II}Fe^{III}$ phase transition. Such ultrafast photoinduced phase transitions induced by a single laser pulse are very promising for optically-driven devices, memories, or actuators exhibiting room temperature switching within a wide thermal regime of bistability between states that can persist long after stimuli. As the streaming powder technique can be easily implemented at X-ray free-electron laser beamlines for femtosecond X-ray studies, we also foresee a broad range of applications to study out-of-equilibrium dynamics for non-reversible phenomena in various functional materials[6,46].

## Methods

### Synthesis and characterization of RbMn$_{0.94}$Co$_{0.06}$Fe

The Rb$_{0.94}$Co$_{0.06}$Mn$_{0.94}$[Fe(CN)$_6$]$_{0.98}$·0.2H$_2$O material was synthesized and characterized by various techniques (Supplementary Methods 1). The crystals are plate-shaped, with good crystallinity and average size of $0.9 \pm 0.3$ μm. Magnetic measurements indicate that the compound exhibits a thermal phase transition between the HT and LT phases with transition temperatures $T_\downarrow = 253$ K and $T_\uparrow = 328$ K. The XRD powder patterns at room temperature and Rietveld analysis correspond for the HT phase to a cubic space group (F$\bar{4}$3m) with a lattice constant of $a_{HT} = 10.5495(6)$ Å and for the LT phase to a tetragonal space group (F$\bar{4}$2m) with $a_{LT} = b_{LT} = 10.0051(11)$ Å and $c_{LT} = 10.4744(16)$ Å. CCDC #2259400 (LT) and #2259401 (HT) contain the supplementary crystallographic data for this paper. These data can be obtained free of charge from The Cambridge Crystallographic Data Centre via www.ccdc.cam.ac.uk/data_request/cif.

### Static photoinduced measurements on RbMn$_{0.94}$Co$_{0.06}$Fe films

RbMn$_{0.94}$Co$_{0.06}$Fe in pure ethanol was deposited on a glass substrate and dried to obtain films of the material (Fig. 1a). Films were cooled below 180 K to reach the $Mn^{III}Fe^{II}$ LT phase and warmed above 350 K to reach the $Mn^{II}Fe^{III}$ HT phase. A 532 nm CW laser (FSDL-532-100T, Frankfurt Laser Company) with 100 mW output power was used to photoexcite during 3 s the LT phase at room temperature ($0.5 \times 0.45$ mm$^2$ spot size (FWHM) at the film position, power density of 120 W cm$^{-2}$) to photoinduce the HT phase. For Fig. 1a, photos were taken with a camera, in the following order and for the same thin film: crystal deposition and cooling at 180 K, photo of the LT phase at room temperature, laser irradiation and photo of the irradiated film at room temperature, heating to 350 K and photo of the heated film.

### Setup for streaming powder diffraction

Time-resolved X-ray diffraction (TR-XRD) measurements were performed on time-resolved beamline ID09 at the European Synchrotron Radiation Facility (ESRF)[53]. The experiment was performed while the ESRF storage ring was operated in 7/8 + 1 filling mode with a single bunch current of 4 mA, corresponding to an X-ray pulse duration of ~35 ps (HWHM). The X-ray central energy was 14.963 keV ($\lambda = 0.824$ Å, 2% bandwidth), and the beam was focused to a spot size of $25 \times 25$ μm$^2$ at the sample position. To perform laser pump–X-ray probe measurements, isolated X-ray pulses are synchronized with a 1 ps laser, whose spot size was 210 μm × 250 μm. The larger laser spot size ensures a homogeneous excitation of the sample in the volume probed by the X-rays. The laser wavelength was set to 650 nm, exciting crystals on the red edge of their absorption spectrum, which results in a penetration depth of about 1.9 μm[54], larger than the average size of the crystals ($0.9 \pm 0.3$ μm). Additionally, we studied how photo-response changes with laser fluence, which varied between 0 and 150 mJ cm$^{-2}$.

For the present streaming powder diffraction study (Supplementary Methods 2–4), crystals of RbMn$_{0.94}$Co$_{0.06}$Fe were dispersed in ethanol and streamed through a free-flowing liquid jet to interact with laser and X-ray pulses. The jet size was 300 μm, and the crystal-solvent weight ratio was 1:270, which corresponds to ~500 crystals per shot in the X-ray focus. The flow rate of the jet was set to 1.2 L h$^{-1}$ (jet velocity of 5 m s$^{-1}$ at the interaction region). As a consequence, a fresh ensemble of crystals, which was not excited by previous laser shots, is interacting with the laser pump–X-ray probe beams. The chosen flow rate limits the observation time window to 10 μs for pump-probe measurements. Jet was circulated in closed loop, and a cooling device was inserted in the liquid circulation loop in order to compensate for the non-reversible photo-conversion of the already-excited crystals (Supplementary Methods 2). X-ray diffraction patterns were recorded in transmission geometry using a Rayonix MX170-HS CCD detector with an integration time of 5 s for each image. The delay between the laser pump pulse and the X-ray probe pulse was varied from −3 ns to +10 μs, and we typically collected 20 diffraction images for each delay. As crystals are randomly oriented in the liquid jet, diffraction images consisted of rings that were azimuthally integrated using pyFAI[68]. We normalized each image over the Q-range of [5.5; 6.0] Å$^{-1}$, where the diffraction signal remains constant along the dynamics.

### XRD analysis

The X-ray scattering patterns of RbMn$_{0.94}$Co$_{0.06}$Fe were obtained by subtracting the background scattering from the solvent. This was done by subtracting the pattern of pure ethanol, followed by polynomial fitting of residual background (Supplementary Methods 3 and 4). Rietveld refinement of the X-ray diffraction data was performed using TOPAS-Academic version 6 software[69], with a fixed wavelength of 0.827389 Å$^{-1}$ (15 keV as calibrated at the ID09 ESRF beamline). For pure LT and HT phases, refinement was done over the scattering vector Q-range [0.9; 5.0] Å$^{-1}$ using the structures found from powder diffraction, and including refinement of lattice parameters and isotropic Debye–Waller thermal factors. Given the symmetry change between tetragonal LT phase and cubic HT phase, refinement was done considering space group F$\bar{4}$2m for LT phase and F$\bar{4}$3m for HT phase in order to directly compare both phases, as explained in our recent work[38]. Sometimes a weak contribution from HT phase was observed in the equilibrium X-ray diffraction data. Analysis of time-resolved and fluence-dependent diffraction patterns was performed following the same procedure as for pure phases. Background subtraction and Rietveld refinement were performed at each delay. More details are provided in Supplementary Methods 3 and 4.

## Data availability

Raw data generated in this study and codes used to process the data have been deposited on the Zenodo repository[70] under https://doi.org/10.5281/zenodo.10227842. Any additional information is available from the corresponding authors upon reasonable request.

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

## Acknowledgements

The authors gratefully acknowledge the Agence Nationale de la Recherche for financial support under grant ANR-19-CE30-0004 ELECTROPHONE, ANR-19-CE07-0027 SMAC, ANR-19-CE29-0018 MULTICROSS and the European Synchrotron Radiation Facility (ESRF) for provision of synchrotron radiation facilities under proposal number CH-6162. G.P. thanks Région Bretagne for partial Ph.D. funding (ARED PHONONIC). I.C. acknowledges financial support from Région Bretagne under grant SAD. E.C. thanks the University of Rennes and the Fondation Rennes 1 for funding. This work was carried out in the frame of a JST FOREST Programme (JPMJFR213Q), a JSPS Grant-in-Aid for Scientific Research (B) (22H02046), Scientific Research (A) (20H00369), and Yasaki Memorial Foundation for Science and Technology. We are grateful to Yugo Nagane (the University of Tokyo) and Mayuko Tanaka (the University of Tsukuba) for their technical support in synthesizing the sample.

## Author contributions

E.C., M.C., H.T. and S.O. conceived and funded the project. M.H., G.P., I.C., C.M., M.L. and Y.W. developed the streaming powder diffraction setup. H.T., S.A. and S.O. synthesized and characterized the sample. M.H., G.P., E.T., S.Z., I.C., C.M., A.V., M.C., M.L. and E.C. performed the time-resolved X-ray diffraction experiment. M.H., G.P., E.T., C.M., M.C. and E.C. analysed the time-resolved X-ray diffraction data. M.H. and R.G.T.R. studied laser-induced sample colour change. M.H. and E.C. set the physical picture with the Landau model. E.C. and M.H. wrote the paper, with contributions from all the authors. All authors have approved the final version of the paper.

## Competing interests

The authors declare no competing interests.
