## [Peer Review File · Nature Communications]

Reviewers' Comments:

Reviewer #1:

Remarks to the Author:

"Ultrafast and persistent photoinduced phase transition at room temperature monitored by streaming powder diffraction" by M.Herve et al. reports the structural dynamics of the photoinduced phase transition (PIPT) in newly designed Cyanide Complexes of the Transition Metals: $\text{Rb}_{0.94}\text{Mn}_{0.94}\text{Co}_{0.06}[\text{Fe}(\text{CN})_6]_{0.98}$ material. They have observed lattice dynamics in quite a wide time range under the one-shot measurement condition to clarify the occurrence of the structural symmetry change and the role of the elastic deformations. In addition, they have reported the excitation intensity dependence of the lattice dynamics and revealed the crossover from the small polaron formation to macroscopic phase changes. They also noted that this phenomenon can be well explained based on the semi-classical Landau framework. This work is the first to demonstrate that the small polaron formation can trigger the macroscopic PIPT as a seed utilizing dynamical structural data. This work will attract a wide field of readers, including basic and materials science. Then referee strongly recommends the publication of this work. However, before publication, the referee considers it necessary to add more explanation about the measurement system. The referee believes that the data on the time dependence (Figs. 2 and 3) are the heart of this work to support the proposed model. Then it is essential to demonstrate that the effect of the crystal flow, especially at a long delay, is small. For example, in the "Setup for Streaming Powder Diffraction" part, the authors comment, "The chosen flow rate limits the observation time window to 10 μs for pump-probe measurements". However, in case the pump light is not the correct top-hat laser beam, considering the speed of the flow (5m/s), during 10 μs , photo excited region (and micro-crystals) moves 50 μm which corresponds to half of the radius (a quarter of diameter) of the excitation pump beam (210x250 μm). Then it is a natural question of the broad field of readers whether the density of the excited crystal is homogeneous or not in the excited pump region considering the flow rate. The referee strongly recommends additional discussion/comment on evaluating the homogeneity of the pump beam in this part or supporting information not to mislead the readers.

Reviewer #2:

Remarks to the Author:

Marius Hervé et coauthors report on the photoinduced phase transition in the material $\text{Rb}_{0.94}\text{Mn}_{0.94}\text{Co}_{0.06}[\text{Fe}(\text{CN})_6]_{0.98}$, from a low-temperature (LT) tetragonal phase to a high-temperature (HT) cubic phase. The material is developed for the present experiment, in order to display a wide thermal hysteresis between the two phases, to be centered around room temperature. The material is photoexcited by ≈ 1 ps laser pulses at 650 nm, and probed by ultrafast x-ray diffraction. To the aim of this experiment, the authors develop a rather interesting streaming powder technique. The experiment proves the possibility to achieve a light-induced phase transition from the LT to the HT phase. Moreover, authors analyze the results of the tetragonal-to-cubic lattice transition using the Landau theory for symmetry-breaking phase transitions.

The experiment is certainly well executed, and the results are presented clearly, providing clear evidence of the validity of the claim. The manuscript is also well written, although in my opinion in some points it could be improved (see in the following).

My main concern is about the novelty of the results discussed, that in my opinion is not sufficient for considering the findings of a broad interest for justifying publication in Nature Communications. While I appreciate the efforts spent in synthesizing a new material that exhibit an (equilibrium) bistability between LT and HT phases across room-temperature, and the efforts spent in developing the interesting setup for streaming-powder-diffraction experiments, the result of a photoinduced switching between a tetragonal LT phase to a HT cubic phase is not new (for example, results are reported in <https://doi.org/10.1016/j.crci.2019.05.002> and <https://doi.org/10.1002/ejic.201801478>). In addition, as far as I can understand, I cannot find specific fingerprints of the PT (photoexcited tetragonal) and PIC (photoinduced cubic) phases revealed by time-resolved diffraction, that differ from those of their equilibrium (LT and HT

respectively) counterpart phases (apart from the fluence-dependent lattice sizes). This makes the x-ray diffraction technique non mandatory for the discussion of the PIPT, once one already knows the system can switch from LT to HT phase, and that, for example, the dielectric function of the material displays dramatic changes in the visible spectral range (I am referring to Fig.1 of Ref.31). Hence a simpler time-resolved transmission experiment could provide better insights into the dynamics with better time resolution and sensitivity.

I also have a few minor points that I think should be addressed for improving the manuscript for publication in a more specialized journal (they are not listed versus their importance):

- 1) I would try to find a 'name'/label' to refer to the new material synthesized, as done in other papers on similar samples, instead of calling it just "(1)"
- 2) Line '27' in the abstract, provides \diamond provide
- 3) If known, I would discuss in the present manuscript the size (radius) of the (small) polarons formed after photoexcitation.
- 4) The 'difference' (in definition) between the PIC and HT and between PT and LT phases becomes more clear only after looking at the sketches reported in Fig. 4 (although as said before I cannot find spectroscopic fingerprints in the diffraction patterns). I would try to anticipate explaining this difference, earlier in the manuscript.
- 5) In Fig. 4, the sketches of the lattice configurations miss the indication of the axes.
- 6) The repetition rate of the experiment (1 kHz) is stated in the main text, but omitted from the Methods section and the SI, where it should also be discussed.
- 7) This number should also be linked more directly to the conclusion (for the used flow rate) that the sample is always 'fresh' every 1 ms and (less obvious) that the limit to the dynamics is 10 μ s.
- 8) The fluence used is extremely high, so I guess (although the pump pulses are 1 ps long) that the numbers provided refer to the incident (rather than to the absorbed) quantity. I am wondering whether, knowing the average crystallite size, it would be possible to provide an absorbed power density, from which the percentage of the excited units could be retrieved.

Reviewer #3:

Remarks to the Author:

The authors are reporting on a single laser shot driven phase transition at room temperature in the newly synthesized compound $\text{Rb}_{0.94}\text{Mn}_{0.94}\text{Co}_{0.06}[\text{Fe}(\text{CN})_6]_{0.98}$. Overcoming a threshold fluence a lattice expansion explained by photoinduced polarons is observed leading to a switching of the material from its low temperature $\text{Mn}^{\text{III}}\text{Fe}^{\text{II}}$ tetragonal ground state into a $\text{Mn}^{\text{II}}\text{Fe}^{\text{III}}$ cubic phase. The change in the lattice structure is permanent. The experimental proof is delivered by means of ultrafast X-ray diffraction using a new developed powder streaming technique.

The article reports three novelties: the synthesis of a new compound with a hysteresis centered around room temperature and hence promising for applications, the development of a streaming powder technique for performing ultrafast XRD measurements and a single laser shot driven phase transition of the synthesized material from its tetragonal low-temperature phase to its cubic high-temperature phase.

The referee is convinced that the developed powder stream technique will be a useful extension of tr-XRD setups and for future use for the community and the novel synthesized material could find interest as well for further studies.

The presented data are interesting and solid, furthermore the authors performed a profound data analysis and developed a model to explain the mechanism behind the single shot phase transition based on former experiments on other materials.

Combining all three novelties, the referee considers the work as adequate for publication in Nature Communications after addressing the below mentioned comments.

- The usage of the word "control" does not seem adequate to the referee since the process is not reversible by optical manners. The referee furthermore would like to ask the authors to comment on the possibility of ultrafast heating instead of the polaron assisted phase transition suggested in

the article. Could the authors comment which kind of charge transfer is supposed to take place and how critical is the excitation wavelength?

- The authors are not very clear on defining the PT and PIC phase in the main text. It would be better to define from the beginning that the PIC phase is described by the HT lattice parameters. Please state more clearly that it is assumed to be identical to the HT phase and label Fig. 3a left part accordingly.

Instead, the variation of the PT phase from the LT and the PIC phase is not getting clear in the manuscript. Please state more clearly where the starting parameters are coming from. What to PT lattice parameters for negative pump-probe delays mean? Isn't it a fully photoinduced phase?

- The time-dependence of the ferroelectric distortion shown on Fig. 2 is not introduced. Does it purely depend on lattice heating due to pumping?

- In line 56 the authors are stating that the new synthesized material presents all required characteristics for an ultrafast photo-switch device and are referring to the supplementary material. However, neither the main text nor the supplementary are explaining how such a device using the new material could look like and what are the exact requirements. The referee proposes to be more explicit.

- In the first part of the manuscript several cross references between main text and supporting information are given, especially regarding the figures. This does influence in a negative way the readability of the article. The referee invites the authors to overthink the structure of Fig. 1. The referee suggests merging Fig. 1 and Fig. S1 from the supplementary. The hysteresis loop should get an extra panel and not be placed arbitrary as an inset of the experimental setup, since it has central importance for understanding the performed experiment. It is also a description missing on how these data are acquired.

Further comment to figure 1: Part a of the figure is not properly described in main text and caption. What kind of images are shown? In which order are the images taken? What are the details about the illumination? Please insert a proper description as you partially did already in the supplementary material.

- In line 129 the authors are stating that 650nm light is promoting the phase transition. Did they try or do they expect a dependence of the excitation wavelength? What would be a threshold energy for inducing the phase transition?

- In line 136 the authors write "The overall global fraction of LT-crystals photo-excited in the streamed suspension is small." Could the authors quantify the fraction? Does it stay constant or increases for each cycle the solution is passing the cooling loop? Since only thermal back-switching is possible, this might be crucial thinking towards device application.

- In line 322 the authors state that the PIC phase is persistent at room-temperature and more generally in a wide thermal hysteretic regime. Did the authors perform temperature dependent measurements? If so, is there a dependence of the threshold fluence on the sample temperature? Is switching to be expected being possible at all at temperatures below 240 K?

- Please state how temperature measurement is done in order to control the temperature of the sample during the experiment in the powder stream apparatus.

The referee is looking forward to seeing this work published after addressing the comments made above.

General formatting: We have modified our paper to fit with Nature Communications formatting instructions: the abstract shortened to less than 150 words, heading sections, formatting, and styles were also checked.

Response to the reviewers' comments

Reviewer #1 (Remarks to the Author):

"Ultrafast and persistent photoinduced phase transition at room temperature monitored by streaming powder diffraction" by M.Herve et al. reports the structural dynamics of the photoinduced phase transition (PIPT) in newly designed Cyanide Complexes of the Transition Metals: $\text{Rb}_{0.94}\text{Mn}_{0.94}\text{Co}_{0.06}[\text{Fe}(\text{CN})_6]_{0.98}$ material. They have observed lattice dynamics in quite a wide time range under the one-shot measurement condition to clarify the occurrence of the structural symmetry change and the role of the elastic deformations. In addition, they have reported the excitation intensity dependence of the lattice dynamics and revealed the crossover from the small polaron formation to macroscopic phase changes. They also noted that this phenomenon can be well explained based on the semi-classical Landau framework. This work is the first to demonstrate that the small polaron formation can trigger the macroscopic PIPT as a seed utilizing dynamical structural data. This work will attract a wide field of readers, including basic and materials science. Then referee strongly recommends the publication of this work.

However, before publication, the referee considers it necessary to add more explanation about the measurement system. The referee believes that the data on the time dependence (Figs. 2 and 3) are the heart of this work to support the proposed model. Then it is essential to demonstrate that the effect of the crystal flow, especially at a long delay, is small. For example, in the "Setup for Streaming Powder Diffraction" part, the authors comment, "The chosen flow rate limits the observation time window to 10 μs for pump-probe measurements". However, in case the pump light is not the correct top-hat laser beam, considering the speed of the flow (5m/s), during 10 μs , photo excited region (and micro-crystals) moves 50 μm which corresponds to half of the radius (a quarter of diameter) of the excitation pump beam (210x250 μm). Then it is a natural question of the broad field of readers whether the density of the excited crystal is homogeneous or not in the excited pump region considering the flow rate. The referee strongly recommends additional discussion/comment on evaluating the homogeneity of the pump beam in this part or supporting information not to mislead the readers.

Response: We would like to thank the reviewer for this comment. Indeed, this is something we checked with great care, but we did not detail the experimental limitations. We therefore include an additional part in the supplementary section 2 to explain why the temporal range we can explore is limited to 10 μs , together with a new Supplementary Figure 4:

See Supplementary Methods 2 and Supplementary fig. 4 for more details:

"The jet velocity... this configuration"

Supplementary Fig. 4. Observation time window in streaming powder diffraction. **a** Schematic representation of excitation profile at different delays, showing the overlap between the laser-irradiated volume (green gaussian profile) and the X-ray probed volume (orange gaussian profile) in the jet, calculated using experimental conditions. **b** Diffraction patterns measured for solution of $\text{RbMn}_{0.94}\text{Co}_{0.06}\text{Fe}$ at different delays (from -3 ns, blue curve, to +800 μs , red curve, see panel **c** for the correspondence between colours and delays) after laser excitation at $62 \text{ mJ}/\text{cm}^2$. The inset shows a zoom around (400) Bragg peaks, where the differential signal with respect to -3 ns is shown. It shows a positive and constant signal around (400)_{PIC} before 10 μs , due to photo-transformation of the crystals, that vanishes after 10 μs , because of loss in laser – X-ray overlap. **c** Integrated intensity of (400)_{PIC} peak as a function of delay (blue curve, integrated over the dashed lines in the inset of **b**, and color-coded dots), together with the modelled laser – X-ray overlap (orange curve). Orange star markers correspond to delays at which the snapshots are represented in panel **a** (1 μs : 0.1% loss of overlap, 6 μs : 5% loss, 10 μs : 10% loss, 22 μs : 50% loss, 100 μs : 100% loss), and orange axis represents the corresponding central position of the photo-excited volume along the jet.

Reviewer #2 (Remarks to the Author):

Marius Hervé et coauthors report on the photoinduced phase transition in the material $\text{Rb}_{0.94}\text{Mn}_{0.94}\text{Co}_{0.06}[\text{Fe}(\text{CN})_6]_{0.98}$, from a low-temperature (LT) tetragonal phase to a high-temperature (HT) cubic phase. The material is developed for the present experiment, in order to display a wide thermal hysteresis between the two phases, to be centered around room temperature. The material is photoexcited by ≈ 1 ps laser pulses at 650 nm, and probed by ultrafast x-ray diffraction. To the aim of this experiment, the authors develop a rather interesting streaming powder technique. The experiment proves the possibility to achieve a light-induced phase transition from the LT to the HT phase. Moreover, authors analyze the results of the tetragonal-to-cubic lattice transition using the Landau theory for symmetry-breaking phase transitions.

The experiment is certainly well executed, and the results are presented clearly, providing clear evidence of the validity of the claim. The manuscript is also well written, although in my opinion in some points it could be improved (see in the following).

My main concern is about the novelty of the results discussed, that in my opinion is not sufficient for considering the findings of a broad interest for justifying publication in Nature Communications. While I appreciate the efforts spent in synthesizing a new material that exhibit an (equilibrium) bistability between LT and HT phases across room-temperature, and the efforts spent in developing the interesting setup for streaming-powder-diffraction experiments, the result of a photoinduced switching between a tetragonal LT phase to a HT cubic phase is not new (for example, results are reported in <https://doi.org/10.1016/j.crci.2019.05.002> and <https://doi.org/10.1002/ejic.201801478>).

Response: The reviewer is mentioning two previous papers: the first one is a broad review about thermal and photoinduced phase transitions in various RbMnFe derivatives and the second one is an X-ray diffraction study of photoswitching inside thermal hysteresis at low temperature. The data presented in these papers are not about time-resolved or out-of-equilibrium structural dynamics. Indeed, the non-reversible photoinduced phase transition within the thermal hysteresis precluded the use of conventional pump-probe techniques on the same sample, which does not relax once excited. We had to develop a new sample streaming technique, which we are introducing in the present paper submitted to Nature Communication, to gain knowledge about out-of-equilibrium dynamics, which was inaccessible by conventional techniques. To our knowledge, this is the first report revealing the out-of-equilibrium structural dynamics, the local and global transient structural deformations, the threshold response from local to macroscopic phase transition and the timescale on which the photoinduced symmetry change from tetragonal to cubic occurs. Photoswitching inside hysteresis was studied in many materials. But this is the first report, to our knowledge, revealing the out-of-equilibrium structural dynamics behind, and performed in a newly synthesized material exhibiting bistability at room temperature, which is very important for applications. We are also convinced that our sample streaming technique will be of interest to study non-reversible phenomena in many materials.

We include a comment about this in the text:

“The present results demonstrate the possibility to induce by a single laser shot an ultrafast PIPT at room temperature, in a functional material designed to exhibit a wide regime of thermal

bistability. Compared to previous single laser shot photoswitching, observed in another RbMnFe derivative at low temperature by static X-ray diffraction studies,⁶⁹ our time-resolved analysis also brings key information on the out-of-equilibrium structural and symmetry dynamics. Indeed, these long-range structural reorganizations can only be directly monitored by ultrafast diffraction techniques.”

In addition, as far as I can understand, I cannot find specific fingerprints of the PT (photoexcited tetragonal) and PIC (photoinduced cubic) phases revealed by time-resolved diffraction, that differ from those of their equilibrium (LT and HT respectively) counterpart phases (apart from the fluence-dependent lattice sizes).

Response: The PIC phase is indeed very similar to the HT phase. And this is important information, since HT and PIC are induced by different external stimuli. Again, since previous studies were not time-resolved, our data shows that the PIC phase, similar to the HT phase is reached within 100 ps.

Regarding the PT phase, this is another story. In the ground LT tetragonal phase, the electronic state is $Mn^{III}Fe^{II}$, with equilibrium lattice parameters $a=b=10.0 \text{ \AA}$ and $c=10.5 \text{ \AA}$. In the photoexcited state, below threshold, the lattice remains tetragonal, but expanded (Fig. 2) due to the formation of CT polaron. Therefore, the photoexcited tetragonal phase (PT) differs from the ground LT tetragonal phase, as the photoinduced CT polarons are responsible for local deformations, monitored through average bond length elongation (Fig. 3).

We rewrote the text to explain these points in more details:

“Fig. 2 shows the evolution of the diffraction pattern of $RbMn_{0.94}Co_{0.06}Fe$ crystals, in the initial LT phase, and measured at room temperature as a function of pump – probe delay t . We used a Rietveld refinement (Supplementary Methods 3) to track the time evolution of the lattice parameters, phase fractions, and bond lengths for the different phases coming into play. At negative delays the diffraction pattern corresponds to the initial $Mn^{III}Fe^{II}$ LT phase (space group), where the characteristic Jahn-Teller ferroelastic distortion translates through the splitting of the $(h00)$ and $(00h)$ Bragg peaks. At positive delays we had to consider two time dependent phases: a tetragonal phase that describes the evolution of the original phase, called hereafter the photoexcited tetragonal (PT) phase, and a photoinduced cubic (PIC) phase, whose crystalline structure is similar to the HT phase.

Fig. 2a shows that for low pumping fluence (15 mJ.cm^{-2}) the initial LT Bragg peaks shift with delay. The structural Rietveld refinements of the diffraction patterns provide the evolution of the lattice parameters and of the ferroelastic distortion $n(t)=(c(t)-a(t))/a_{HT}$ for each individual delay t .”

This makes the x-ray diffraction technique non mandatory for the discussion of the PIPT, once one already knows the system can switch from LT to HT phase, and that, for example, the dielectric function of the material displays dramatic changes in the visible spectral range (I am referring to Fig.1 of Ref.31). Hence a simpler time-resolved transmission experiment could provide better insights into the dynamics with better time resolution and sensitivity.

Response: Indeed, we also wanted to perform optical spectroscopy studies, as suggested by the reviewer. But here again the conventional time-resolved transmission experiments could not be performed to study the out-of equilibrium dynamics during thermal hysteresis, because of the non-reversible nature of the PIPT. The sample streaming technique we used is required

to refresh the sample before exciting it anew in pump-probe studies. However, given that we use a powder of about 1 μm size crystals in the stream, optical technique could not be used due to important scattering. In addition, optical spectroscopy provides no information about symmetry change or volume expansion, which are at the heart of the physics behind elastically-driven cooperative PIPT. It was therefore mandatory to use time-resolved X-ray diffraction with sample streaming technique to get new insight into the out-of-equilibrium dynamics behind PIPT within thermal hysteresis.

We comment about this in the conclusion:

“Compared to previous single laser shot photoswitching, observed in another RbMnFe derivative at low temperature by static X-ray diffraction studies,⁶⁹ our time-resolved analysis also brings key information on the out-of-equilibrium structural and symmetry dynamics. Indeed, these long-range structural reorganizations can only be directly monitored by ultrafast diffraction techniques.”

I also have a few minor points that I think should be addressed for improving the manuscript for publication in a more specialized journal (they are not listed versus their importance):

1) I would try to find a ‘name’/‘label’ to refer to the new material synthesized, as done in other papers on similar samples, instead of calling it just “(1)”

We thank the reviewer for his suggestion. We changed (1) to **RbMn_{0.94}Co_{0.06}Fe**.

2) Line ‘27’ in the abstract, provides provide

We thank the reviewer for careful reading.

3) If known, I would discuss in the present manuscript the size (radius) of the (small) polarons formed after photoexcitation.

The size of the CT polaron is known from previous IR spectroscopy studies, which revealed that in addition to the characteristic decrease of the ground CoIII-N-C-FeII band and appearance of the CoII-N-C-FeIII band due to CT, new IR bands associated with CoII-N-C-FeII and CoIII-N-C-FeIII modes appear, which is characteristic of the localization of CT polaron at the level of a single Co-N-C-Fe unit [Ref Asahara PHYSICAL REVIEW B 86, 195138 (2012)].”

We add a comment about this in the text.

“which are localized at the level of a single Co-N-C-Fe unit as previously characterized by time-resolved IR spectroscopy.⁴⁴”

4) The ‘difference’ (in definition) between the PIC and HT and between PT and LT phases becomes more clear only after looking at the sketches reported in Fig. 4 (although as said before I cannot find spectroscopic fingerprints in the diffraction patterns). I would try to anticipate explaining this difference, earlier in the manuscript.

We rewrote the text to explain these points in more details (see also response to the comment above regarding PIC and PT phases):

“Fig. 2 shows the evolution of the diffraction pattern of RbMn_{0.94}Co_{0.06}Fe crystals, in the initial LT phase, and measured at room temperature as a function of pump – probe delay t . We used a Rietveld refinement (Supplementary Methods 3) to track the time evolution of the lattice parameters, phase fractions, and bond lengths for the different phases coming into play. At negative delays the diffraction pattern corresponds to the initial Mn^{III}Fe^{II} LT phase (space group), where the characteristic Jahn-Teller ferroelastic distortion translates through the

splitting of the (*h00*) and (*00h*) Bragg peaks. At positive delays we had to consider two time dependent phases: a tetragonal phase that describes the evolution of the original phase, called hereafter the photoexcited tetragonal (PT) phase, and a photoinduced cubic (PIC) phase, whose crystalline structure is similar to the HT phase.

Fig. 2a shows that for low pumping fluence (15 mJ.cm^{-2}) the initial LT Bragg peaks shift with delay. The structural Rietveld refinements of the diffraction patterns provide the evolution of the lattice parameters and of the ferroelastic distortion $n(t)=(c(t)-a(t))/a_{\text{HT}}$ for each individual delay *t*.”

5) In Fig. 4, the sketches of the lattice configurations miss the indication of the axes. This is a nice suggestion, Fig. 4 was modified accordingly.

6) The repetition rate of the experiment (1 kHz) is stated in the main text, but omitted from the Methods section and the SI, where it should also be discussed.

7) This number should also be linked more directly to the conclusion (for the used flow rate) that the sample is always ‘fresh’ every 1 ms and (less obvious) that the limit to the dynamics is 10 us.

This is discussed in detail in the Supplementary Methods 2 (see also response to reviewer 1 and Supplementary Figure 4)

”The jet velocity was set to 5 m.s^{-1} at the interaction region, which corresponds to a flow rate of 1.2 L/h, i.e. $0.33 \text{ mm}^3/\text{ms}$. Given the repetition rate and laser spot size, the laser beam irradiates a volume of 0.016 mm^3 ($210 \mu\text{m} \times 250 \mu\text{m} \times 300 \mu\text{m}$) every 1 ms. Thus, among the solution that flow through the jet nozzle, a fraction of 5 % only is irradiated by the laser. Additionally, a single photoexcited volume moves 5 mm downstream before the next laser shot arrives (1 ms later), which is lower than the vertical laser spot size ($210 \mu\text{m}$). This ensures that a renewed ensemble of crystals, i.e., which was not excited by the previous laser shot, is interacting with each laser pump – X-ray probe pulses duo.”

8) The fluence used is extremely high, so I guess (although the pump pulses are 1 ps long) that the numbers provided refer to the incident (rather than to the absorbed) quantity. I am wondering whether, knowing the average crystallite size, it would be possible to provide an absorbed power density, from which the percentage of the excited units could be retrieved. Indeed, numbers refer to the incident fluence. Given that crystals are dispersed in solution, it is hard to estimate precisely the number of absorbed photons. What we know from previous studies is that quantum efficiency is of the order of unity and that volume changes linearly with CT fraction. If we compare the relative expansion shown in Fig3b at 62 mJ.cm^{-2} ($DV=0.01$) to the one of the LT to HT phase transition ($DV=0.1$), we estimate the percentage of photoexcited units to 10%.

We mention this in the text

“Since is hard to estimate precisely the number of absorbed photons, we estimate percentage of photoexcited units by considering that quantum efficiency is of the order of unity and volume changes linearly with CT fraction.^{34,35} Compared to the relative volume expansion from LT to HT phases ($DV=0.1$) the relative photoexpansion of the PT phase at 62 mJ.cm^{-2} ($DV=0.01$, in Fig. 3b) allows to estimate that 10% of the MnFe units are photoexcited around excitation threshold.”

Reviewer #3 (Remarks to the Author):

The authors are reporting on a single laser shot driven phase transition at room temperature in the newly synthesized compound $\text{Rb}_{0.94}\text{Mn}_{0.94}\text{Co}_{0.06}[\text{Fe}(\text{CN})_6]_{0.98}$. Overcoming a threshold fluence a lattice expansion explained by photoinduced polarons is observed leading to a switching of the material from its low temperature $\text{Mn}^{\text{III}}\text{Fe}^{\text{II}}$ tetragonal ground state into a $\text{Mn}^{\text{II}}\text{Fe}^{\text{III}}$ cubic phase. The change in the lattice structure is permanent. The experimental proof is delivered by means of ultrafast X-ray diffraction using a new developed powder streaming technique.

The article reports three novelties: the synthesis of a new compound with a hysteresis centered around room temperature and hence promising for applications, the development of a streaming powder technique for performing ultrafast XRD measurements and a single laser shot driven phase transition of the synthesized material from its tetragonal low-temperature phase to its cubic high-temperature phase.

The referee is convinced that the developed powder stream technique will be a useful extension of tr-XRD setups and for future use for the community and the novel synthesized material could find interest as well for further studies.

The presented data are interesting and solid, furthermore the authors performed a profound data analysis and developed a model to explain the mechanism behind the single shot phase transition based on former experiments on other materials.

Combining all three novelties, the referee considers the work as adequate for publication in Nature Communications after addressing the below mentioned comments.

- The usage of the word “control” does not seem adequate to the referee since the process is not reversible by optical manners.

Response: Taking into account the remark from the reviewer, we changed the sentence “The present results demonstrate that we reached single laser shot and ultrafast PIPT control at room temperature in a functional material designed to exhibit wide regime of thermal bistability.”

to

“The present results demonstrate the possibility to induce by a single laser shot an ultrafast PIPT at room temperature, in a functional material designed to exhibit a wide regime of thermal bistability.”

We also rewrote “a volume expansion controlled by the long-lived photo-induced” as

“a volume expansion induced by the long-lived photo-induced”

We used “control” in the introduction, which is more general to challenges in material's science and PIPT phenomena.

The referee furthermore would like to ask the authors to comment on the possibility of ultrafast heating instead of the polaron assisted phase transition suggested in the article.

Response: This is a very interesting point to discuss. We briefly mentioned in the text page 8

“Since RbMnFe and derivative materials exhibit no thermal expansion,⁵⁷ it was shown that such long-range lattice deformation results from the structural trapping of photoinduced Mn^{II}Fe^{III} CT state of higher volume, which forms within 200 fs and decays within 10 μ s.³⁵”
To answer the reviewer, we develop this point in the discussion part.

“This timescale matches the ratio between crystals’ radius (450 nm) and sound velocity ($c=4300 \text{ m}\cdot\text{s}^{-1}$ 58), which is characteristic of an elastically-driven process.^{34,59}”

Indeed, since RbMnFe derivatives exhibit no thermal expansion,⁵⁷ the initial lattice expansion or ferroelastic distortion cannot be explained by a simple laser heating of the lattice”

Could the authors comment which kind of charge transfer is supposed to take place and how critical is the excitation wavelength?

We have included a comment about this in the text:

“The photoinduced $\text{Mn}^{\text{III}}\text{Fe}^{\text{II}} \rightarrow \text{Mn}^{\text{II}}\text{Fe}^{\text{III}}$ CT occurs within 200 fs over a broad excitation spectral range including Fe to Mn CT band (around 500 nm) and Mn-centred band (around 650 nm).³¹ We excited the sample at 650 nm, which is a good compromise between excitation efficiency and penetration depth (of the order of sample thickness).”

Fig. 1 was also modified to describe the charge-transfer process:

- The authors are not very clear on defining the PT and PIC phase in the main text. It would be better to define from the beginning that the PIC phase is described by the HT lattice parameters. Please state more clearly that it is assumed to be identical to the HT phase and label Fig. 3a left part accordingly.

Instead, the variation of the PT phase from the LT and the PIC phase is not getting clear in the manuscript. Please state more clearly where the starting parameters are coming from.

What to PT lattice parameters for negative pump-probe delays mean? Isn't it a fully photoinduced phase?

Response: These points were also raised by reviewer 2 (See comments above for modifications).

“Fig. 2 shows the evolution of the diffraction pattern of $\text{RbMn}_{0.94}\text{Co}_{0.06}\text{Fe}$ crystals, in the initial LT phase, and measured at room temperature as a function of pump – probe delay t . We used a Rietveld refinement (Supplementary Methods 3) to track the time evolution of the lattice parameters, phase fractions, and bond lengths for the different phases coming into play. At negative delays the diffraction pattern corresponds to the initial $\text{Mn}^{\text{III}}\text{Fe}^{\text{II}}$ LT phase (space group), where the characteristic Jahn-Teller ferroelastic distortion translates through the splitting of the $(h00)$ and $(00h)$ Bragg peaks. At positive delays we had to consider two time dependent phases: a tetragonal phase that describes the evolution of the original phase, called hereafter the photoexcited tetragonal (PT) phase, and a photoinduced cubic (PIC) phase, whose crystalline structure is similar to the HT phase.

Fig. 2a shows that for low pumping fluence ($15 \text{ mJ}\cdot\text{cm}^{-2}$) the initial LT Bragg peaks shift with delay. The structural Rietveld refinements of the diffraction patterns provide the evolution of the lattice parameters and of the ferroelastic distortion $n(t)=(c(t)-a(t))/a_{\text{HT}}$ for each individual delay t .”

These points were also raised by reviewer 2.

In addition, Fig. 3a was labeled correctly, by replacing “Tetragonal LT phase” with “Tetragonal (PT) phase”

- The time-dependence of the ferroelectric distortion shown on Fig. 2 is not introduced.

Does it purely depend on lattice heating due to pumping?

Response: We explain more clearly in the text that

“The structural Rietveld refinements of the diffraction patterns provide the evolution of the lattice parameters and of the ferroelastic distortion $n(t)=(c(t)-a(t))/a_{HT}$ for each individual delay t_i ”.

As explained above, the lattice parameters do not exhibit thermal lattice expansion. Therefore is also driven by elastic interaction and the coupling of the polaron with crystalline lattice as explained now in the text.

“Indeed, since RbMnFe derivatives exhibit no thermal expansion,⁵⁷ the initial lattice expansion or ferroelastic distortion cannot be explained by a simple laser heating of the lattice.”

- In line 56 the authors are stating that the new synthesized material presents all required characteristics for an ultrafast photo-switch device and are referring to the supplementary material. However, neither the main text nor the supplementary are explaining how such a device using the new material could look like and what are the exact requirements. The referee proposes to be more explicit.

Response: Indeed, the way this sentence was written was misleading, as the reference to supplementary information concerned sample characterization and not photo-switching device characteristic. We rewrote the sentences to avoid confusion:

“Here we present a newly synthesized material of this class, $Rb_{0.94}Mn_{0.94}Co_{0.06}[Fe(CN)_6]_{0.98} \cdot 0.2H_2O$ ($RbMn_{0.94}Co_{0.06}Fe$), which exhibits a wide thermal hysteresis centred at room temperature (Supplementary Methods 1) resulting from coupled CT and symmetry-breaking (SB).”

The characteristics required characteristics are mentioned in the introduction

“For optically-driven photonic devices, memories, or actuators, the photo-response must combine important characteristics such as room temperature switching, wide thermal regime of bistability, photoinduced states that can persist long after stimuli, single laser shot with threshold switching and ultrafast dynamics”

and is also underlined in the conclusion

“Such ultrafast photoinduced phase transitions induced by a single laser pulse are very promising for optically-driven devices, memories, or actuators exhibiting room temperature switching within a wide thermal regime of bistability between states that can persist long after stimuli.”

- In the first part of the manuscript several cross references between main text and supporting information are given, especially regarding the figures. This does influence in a negative way the readability of the article.

Response: Following Nature Communication’s style, we reduced the number of references to supplementary figures, by making more global reference to supplementary methods or discussions.

The referee invites the authors to overthink the structure of Fig. 1. The referee suggests merging Fig. 1 and Fig. S1 from the supplementary. The hysteresis loop should get an extra

panel and not be placed arbitrary as an inset of the experimental setup, since it has central importance for understanding the performed experiment. It is also a description missing on how these data are acquired. Further comment to figure 1: Part a of the figure is not properly described in main text and caption. What kind of images are shown? In which order are the images taken? What are the details about the illumination? Please insert a proper description as you partially did already in the supplementary material.

Response: As suggested by the reviewer, we changed Fig. 1 and took this opportunity to explain better the charge-transfer process. We also improved the caption of Fig. 1.

- In line 129 the authors are stating that 650nm light is promoting the phase transition. Did they try or do they expect a dependence of the excitation wavelength? What would be a threshold energy for inducing the phase transition?

Response: As indicated in the answer above “The photoinduced $\text{Mn}^{\text{III}}\text{Fe}^{\text{II}} \rightarrow \text{Mn}^{\text{II}}\text{Fe}^{\text{III}}$ CT occurs within 200 fs over a broad excitation spectral range including Fe to Mn CT band (around 500 nm) and Mn-centred band (around 650 nm).³¹ We excited the sample at 650 nm, which is a good compromise between excitation efficiency and penetration depth (of the order of sample thickness).” We could not perform measurements for different laser wavelengths due to time limitation.

- In line 136 the authors write “The overall global fraction of LT-crystals photo-excited in the streamed suspension is small.” Could the authors quantify the fraction? Does it stay constant or increases for each cycle the solution is passing the cooling loop? Since only thermal back-switching is possible, this might be crucial thinking towards device application.

Response: This sentence was unclear and confusing. We were not talking about the fraction of crystals in the laser beam, which are excited, but of the fraction of crystals in the stream, which are exposed to laser shots. We changed the sentence to

“The overall global fraction of LT crystals exposed to laser shot in the streamed suspension is small” and rewrote the corresponding part of Supplementary Methods 2 & 3.

- In line 322 the authors state that the PIC phase is persistent at room-temperature and more generally in a wide thermal hysteretic regime. Did the authors perform temperature dependent measurements? If so, is there a dependence of the threshold fluence on the sample temperature? Is switching to be expected being possible at all at temperatures below 240 K? The reviewer is asking a very interesting question indeed. This was not possible during our experiment at ESRF, as the sample streaming device was a prototype. We are collaborating with ESRF to develop a streaming device with temperature control at the position of the stream. Based on our results presented here, we also wanted to learn more and we wrote a proposal for the ESRF for tracking if threshold fluence depends on temperature. We have obtained beamtime for performing such an experiment in February. As data analysis is complex and takes time, we hope to answer this question during the summer of 2024, if the experiment is successful .

- Please state how temperature measurement is done in order to control the temperature of the sample during the experiment in the powder stream apparatus.

This is explained in details in Supplementary Methods 2

“The cooling temperature, measured using a thermocouple at the end of the heat exchanger, was controlled using a PID feedback loop on the flow rate of cooling nitrogen gas. Following passage through the cooling device, the solution flows through a 2 m tube at room temperature before the jet nozzle, ensuring that the solution is brought back to room temperature at the interaction region. This was checked by measuring the temperature of the solution at the jet position, using a thermocouple. Additionally, temperature of the PBA reservoir was also checked from times to times.”

The referee is looking forward to seeing this work published after addressing the comments made above.

We thank the reviewers for the very interesting and constructive remarks.

Reviewers' Comments:

Reviewer #1:

Remarks to the Author:

"Ultrafast and persistent photo-induced phase transition at room temperature monitored by streaming powder diffraction" by M.Herve et al. reports the structural dynamics of the photo-induced phase transition (PIPT) in newly designed Cyanide Complexes of the Transition Metals: $\text{Rb}_{0.94}\text{Mn}_{0.94}\text{Co}_{0.06}[\text{Fe}(\text{CN})_6]_{0.98}$ material. The revised manuscript is well constructed, and all my previous questions are well explained. Of course, works on similar materials and parts of the photo-induced effects have been reported previously. However, observed lattice dynamics in quite a wide time range under the one-shot measurement condition have clarified the total dynamical view of the structural symmetry change and the role of the elastic deformations for the first time. This work demonstrates the importance of the broad time range observation for comprehensively understanding the photo-induced structural phase transition, which has become possible due to the progress in experimental technique. Then, the referee strongly recommends the publication of this work.

Reviewer #3:

Remarks to the Author:

I am appreciating the effort the authors did for revising the manuscript. I do not have any further concerns for publication and recommend accepting the manuscript in the present form.

Dear editor,

We are happy to see that the reviewers appreciate the modifications made to the paper and recommend publication of our work in the present form.

Since they don't have additional remarks or comments we did not change the manuscript further.

Best wishes and thank you again for taking care of our paper.

Best wishes,

Eric Collet, for the authors.

REVIEWERS' COMMENTS

Reviewer #1 (Remarks to the Author):

"Ultrafast and persistent photo-induced phase transition at room temperature monitored by streaming powder diffraction" by M.Herve et al. reports the structural dynamics of the photo-induced phase transition (PIPT) in newly designed Cyanide Complexes of the Transition Metals: $\text{Rb}_{0.94}\text{Mn}_{0.94}\text{Co}_{0.06}[\text{Fe}(\text{CN})_6]_{0.98}$ material. The revised manuscript is well constructed, and all my previous questions are well explained. Of course, works on similar materials and parts of the photo-induced effects have been reported previously. However, observed lattice dynamics in quite a wide time range under the one-shot measurement condition have clarified the total dynamical view of the structural symmetry change and the role of the elastic deformations for the first time. This work demonstrates the importance of the broad time range observation for comprehensively understanding the photo-induced structural phase transition, which has become possible due to the progress in experimental technique. Then, the referee strongly recommends the publication of this work.

Reviewer #3 (Remarks to the Author):

I am appreciating the effort the authors did for revising the manuscript. I do not have any further concerns for publication and recommend accepting the manuscript in the present form.